# Syntopy between Endangered San Joaquin Kit Foxes and Potential Competitors in an Urban Environment

**DOI:** 10.3390/ani13203210

**Published:** 2023-10-14

**Authors:** Brian L. Cypher, Nicole A. Deatherage, Erica C. Kelly, Tory L. Westall

**Affiliations:** Endangered Species Recovery Program, California State University-Stanislaus, 1 University Circle, Turlock, CA 95382, USA; ndeatherage@esrp.csustan.edu (N.A.D.); ekelly@esrp.csustan.edu (E.C.K.); twestall@esrp.csustan.edu (T.L.W.)

**Keywords:** Coexistence, competition, endangered species, mesocarnivores, San Joaquin kit fox, syntopy, urban environment, *Vulpes macrotis mutica*

## Abstract

**Simple Summary:**

The endangered San Joaquin kit fox (*Vulpes macrotis mutica*; SJKF) occurs in the city of Bakersfield, CA, along with other species such as domestic cats (*Felis catus*), striped skunks (*Mephitis mephitis*), raccoons (*Procyon lotor*), and opossums (*Didephis virginiana*). We used data from camera stations deployed throughout the city during 2015–2022 to assess whether the other species compete with SJKF or coexist syntopically (i.e., occur in the same habitats without competing). Detections of the other species were not associated with those of SJKF, either in areas of high, medium, or low suitability for SJKF or for all areas combined. Also, the abundance of the other species did not increase when SJKF declined from a sarcoptic mange epidemic. The other species were detected at camera stations, with SJKF indicating that they used the same areas. Competition between the SJKF and the other species may be negligible, likely due to high resource abundance. Thus, SJKF and the other species appear to coexist syntopically in the urban environment. This additional SJKF population contributes to the conservation and recovery of this endangered species.

**Abstract:**

The endangered San Joaquin kit fox (*Vulpes macrotis mutica*; SJKF) occurs in the city of Bakersfield, CA, where several putative competitors also occur, including domestic cats (*Felis catus*), striped skunks (*Mephitis mephitis*), raccoons (*Procyon lotor*), and opossums (*Didephis virginiana*). We used data from a multi-year (2015–2022) city-wide camera station survey to assess whether the other species were simply sympatric with SJKF or coexisting syntopically (i.e., occurring in the same habitats without apparent competition). Annual detection rates for the other species were not correlated with SJKF rates either within SJKF habitat suitability categories (low, medium, and high) or for all categories combined. Also, detection rates for the other species did not increase in response to a significant decline in SJKF abundance caused by sarcoptic mange. The use of all SJKF habitat suitability categories by the other species and co-detections with SJKF at camera stations indicate high spatial overlap. Interference and exploitative competition between the species are apparently negligible, likely due to similar body sizes and high resource abundance. Thus, SJKF and the other species appear to be coexisting syntopically in the urban environment, resulting in a significant additional SJKF population that facilitates range-wide conservation and recovery of this endangered species.

## 1. Introduction

The San Joaquin kit fox (*Vulpes macrotis mutica*; SJKF) is a small canid endemic to the San Joaquin Desert region in central California, USA [1,2]. The SJKF was once widely distributed throughout this region in arid shrubland and grassland habitats [1,3]. Considerable habitat within the range of the SJKF has been converted to agricultural, urban, and industrial uses [3,4]. Due to this profound habitat loss, the SJKF was listed as federally endangered in 1967 and California threatened in 1980 [1]. It currently persists in a metapopulation consisting of three “core” populations and less than a dozen smaller “satellite” populations [3,5]. Range-wide, San Joaquin kit foxes likely number less than 5000 and possibly fewer [3,6]. Thus, all remaining populations are considered critical for the conservation and recovery of this species.

Paradoxically, although urban development is one of the primary causes of habitat destruction in the San Joaquin Desert, SJKF occurs in the city of Bakersfield [7,8]. SJKF are widespread in the urban environment and are commonly observed using a diversity of land uses, particularly campuses (e.g., schools, churches), maintained open space (e.g., parks, golf courses), low- to medium-density residential areas (e.g., apartments, nursing homes), commercial areas, and undeveloped lots [6,7,9]. Cypher et al. [6] used occupancy analysis to classify urban habitat suitability and then determined that 121 km^2^ of high suitability habitat, 196 km^2^ of medium suitability habitat, and 40 km^2^ of low suitability habitat were present in Bakersfield. 

Multiple potential competitors with SJKF are also present in Bakersfield [7,10] and include domestic cats (*Felis catus*), domestic dogs (*Canis lupus familiaris*), striped skunks (*Mephitis mephitis*), opossums (*Didephis virginiana*), raccoons (*Procyon lotor*), red foxes (*Vulpes vulpes*), gray foxes (*Urocyon cineareoargenteus*), and coyotes (*Canis latrans*). These species are clearly sympatric (occur in the same region) with SJKF in Bakersfield [10,11] but it is unclear to what extent they are syntopic (coexist in the same habitat). Of particular interest are the species that are similar in body size to SJKF because of the potential for greater niche overlap and, consequentially, greater resource competition [12,13,14]. Thus, syntopy between SJKF and other species would imply the presence of habitat conditions or differences in life-history attributes that facilitate coexistence. Lack of syntopy would imply that competition may occur between species, with sympatry being maintained through some mitigating factor such as spatial, temporal, or resource partitioning. 

Our objective in this study was to determine whether SJKF occurs syntopically in an urban environment with four similar-sized sympatric species: domestic cats, striped skunks, raccoons, and opossums. Specifically, we tested for differences in detection rates at camera stations with and without SJKF detections as well as changes in rates relative to annual SJKF abundance to identify any interspecific avoidance by SJKF of the other species. Any avoidance would indicate a lack of syntopy. Habitat attributes could also influence the degree of syntopy between species. Thus, we assessed the degree of syntopy between SJKF and each species relative to habitat suitability for SJKF. We focused on the suitability of the habitat for SJKF because this species is endangered, and we wanted to gain a better understanding of factors influencing its distribution and abundance in the urban environment, including the potential for competition and the implications for SJKF conservation.

## 2. Materials and Methods

### 2.1. Study Area

Bakersfield is located in Kern County in the southern San Joaquin Valley in central California, USA (Figure 1). As of 2020, the area of the city was 388 km^2^, and the human population was ca. 391,438 [15]. The terrain in the region is relatively flat, with an average elevation of 124 m. The climate in the region is characterized by hot, dry summers and cool winters, with precipitation primarily being received in the winter. The average high and low temperatures are 13.7 °C and 3.9 °C in December and 36.2 °C and 21.4 °C in July, and the mean annual precipitation is 164 mm [16]. Occupied SJKF habitat borders Bakersfield to the northeast and southwest [3], with irrigated agriculture bordering the city elsewhere.

### 2.2. Data Collection and Analysis

We used occurrence data gathered from 2015 to 2022 during annual surveys conducted in Bakersfield to assess the frequency and distribution of SJKF with sarcoptic mange [17]. The surveys were conducted using camera stations, and other species were also detected during the surveys. These data were used to infer syntopy between SJKF and other species. These surveys are described in detail in Deatherage et al. [9] and Cypher et al. [3]. In brief, a sampling grid consisting of 357 1-km^2^ cells was digitally overlaid on a map of the city. Using the randomization function in Excel 2010 (Microsoft, Redmond, WA), we selected a subset of the cells to survey. Annually, we sampled 111 grid cells (31% of the total), and the same cells were sampled each year to eliminate inter-annual spatial variation as a confounding variable (Figure 1). Within each selected grid cell, we then identified locations (1) that were accessible to SJKF and the other species and (2) where the risk of camera theft was low (i.e., locations with restricted public access or where a camera could be placed in a cryptic location). Consequently, most camera stations were placed in locations such as school campuses, city or county storm water drainage basins, municipal facilities, churches, golf courses, private businesses (with owner permission), and undeveloped parcels. The estimated mean home range size for SJKF in Bakersfield is 0.78 km^2^ [6], which results in an estimated home range diameter of 1 km. We attempted to find locations toward the center of cells in which to place cameras such that stations in adjacent cells would be separated by at least 1 km.

Within each sampled grid cell, we employed an automated camera station and used a methodology specifically developed to survey for SJKF and other carnivores [18]. We used Cuddeback Digital Black Flash IR cameras (Model 1255, Non Typical Inc., Green Bay, WI) that take high-resolution images (20 megapixels) and that employ a “black flash” infrared LED flash that is not visible to humans. Usually, the cameras were secured to 1.2 m U-posts using zip-ties. At locations where cameras might be more easily discovered by the public, we placed the cameras in protective cases (“CuddeSafe” Model 3327, Non Typical Inc., Green Bay, WI) that were secured with a cable lock to fences, trees, or other immobile structures. To attract SJKF and other mesocarnivores to the camera stations, several drops of a scent lure (Carman’s Canine Call Lure, New Milford, PA) were placed in front of the camera and on surrounding vegetation. A 163 ml can of cat food was staked to the ground approximately 2 m in front of each camera using 30 cm nails, and the cans were perforated to allow scent to void. Animals could not access the food in the can. Because 97.1% of the first detections of a SJKF at camera stations typically occur within six nights [18], cameras were deployed at each location for seven nights. We felt that the station spacing and duration employed provided ample opportunity to detect SJKF and other mesocarnivores using a given cell. We did not have comparable information on time to detection for the other species, but our experience in analyzing camera images over the multi-year study suggested that seven nights was sufficient for detecting the other species at a given location. Images were then downloaded from each camera and examined to determine whether SJKF or other species had visited each station during the seven days it was deployed each year.

The suitability of the habitat in each cell for SJKF had previously been determined using occupancy analysis and categorized as low, medium, or high [6]. High-suitability habitat generally had a higher proportion of campuses, low- to medium-density residential areas, and open green space such as parks, golf courses, and cemeteries. Low-suitability habitat generally had a higher proportion of roads. The number of surveyed cells in each suitability category was approximately proportional to the suitability composition across the entire 357-km^2^ study area. Consequently, the surveyed cells included 12 of low suitability, 64 of medium suitability, and 35 of high suitability.

For each year that surveys were conducted and for each species, the number of detections was determined across all cells as well as for each SJKF habitat suitability category. For each year and species, a detection rate was derived by dividing the number of detections by the number of surveyed cells. For each species, a mean annual rate was calculated for all surveyed cells and for each SJKF habitat suitability category. Trends in annual detection rates for SJKF were compared to those for each of the other species using Pearson correlation analysis. This was conducted for the entire study area and also for each habitat suitability category. To determine whether detection rates varied among habitat suitability categories for each species, mean annual detection rates were compared between categories using a one-way analysis of covariance with a Bonferroni post hoc test for pair-wise comparisons. The number of annual detections for each species was used as a covariate in the models to account for any changes in abundance over time.

To determine whether use of SJKF habitat suitability categories differed between SJKF and other species, mean detection rates were compared between SJKF and each of the other species using a two-way analysis of covariance conducted within a general linear model framework. The number of annual detections for each species was used as a covariate in the models to account for any changes in abundance over time and also for differences in abundance between species. Pair-wise comparisons between habitat suitability categories were conducted using a Bonferroni post hoc test.

Spatial overlap between SJKF and other species was assessed by examining the detection rates of other species in grid cells in which SJKF had also been detected. In these cells, the annual detection rate was determined for each species in each SJKF habitat suitability category. For each species, the mean annual rate was compared among categories using a one-way analysis of variance with a Tukey post hoc test for pair-wise comparisons.

An arcsine transformation was conducted on detection rates prior to analyses to normalize the data [19]. All statistical tests were conducted in SPSS (SPSS Statistics package, ver. 29.0.1.1; IBM, Armonk, NY, USA). For all statistical analyses, we set *α* at 0.10. We chose a more relaxed *α* value to reduce the risk of committing a Type II error [20]. Detecting trends with ecological data can be challenging because all potential confounding factors cannot be controlled [21]. By reducing the Type II error rate, we were more likely to detect potential relationships that could be further investigated [22,23,24,25,26].

## 3. Results

During 2015–2022, 105 to 111 camera stations were operated each year for a total of 865 week-long sessions. Domestic cats were the most frequently detected species, followed by SJKF, opossum, striped skunk, and raccoon (Table 1). SJKF had a relatively wide range of detection rates due to a population decline attributable to a sarcoptic mange epidemic (Figure 2). Annual detection trends for the other species were not significantly correlated with SJKF trends (Table 1). When examined by habitat suitability category (Figure 3), annual detection trends were also not significantly correlated with SJKF trends in any of the categories (low: *r* = −0.06 to 0.29; medium: *r* = −0.16 to 0.29; high: *r* = −0.16 to 0.17; *p* > 0.1 for all correlations).

Mean annual detection rates varied among SJKF habitat suitability categories for all species (Table 2). Patterns of detection in suitability categories were similar between SJKF and raccoons, with rates being highest in high suitability areas and lowest in low suitability areas (Figure 4). Interestingly, detection rates were lowest in medium-suitability areas for cats, striped skunks, and opossums. When adjusted for differences in annual detection rates through analysis of co-variance, detection rates of SJKF and raccoons among suitability categories were similar (*F*_2,41_ = 2.00, *p* = 0.149), but rates among categories differed between SJKF and cats (*F*_2,41_ = 11.96, *p* < 0.001), striped skunks (*F*_2,41_ = 15.14, *p* < 0.001), and opossums (*F*_2,41_ = 5.01, *p* = 0.011).

At locations with SJKF detections, mean ± SE annual detection rates for other species were 75.0 ± 4.0 for cats, 11.8 ± 2.1 for striped skunks, 11.7 ± 4.2 for raccoons, 10.8 ± 2.0 for opossums, and 80.8 ± 17.8 for all species combined. Detection rates for other species at stations where SJKF were also detected varied among SJKF habitat suitability categories for all species except raccoons (Table 3). All of the species were detected at stations with SJKF in all of the habitat categories except for opossums, which were not detected at stations with SJKF in low-suitability areas. Cats were detected at stations with SJKF at higher rates in low-suitability areas and lower rates in high- and medium-suitability areas. Conversely, striped skunks and opossums were detected at stations with SJKF at higher rates in high-suitability areas and lower rates in medium- and low-suitability areas.

## 4. Discussion

Competitive interactions, either interference or exploitative, are common where carnivore species are sympatric, and these interactions have a significant role in the composition and structure of the carnivore community in a given location [27,28,29,30]. Such competitive pressures can be particularly pronounced between similar-sized species [12,13,31]. Furthermore, in urban environments, competition may be enhanced or mediated by anthropogenic influences including novel foods, food subsidies (e.g., wildlife feeding, pet food, discarded food), novel predators (e.g., domestic dogs and cats), and novel mortality factors (e.g., vehicle traffic) [32,33,34]. Consequently, some species may be suppressed or even excluded from urban environments by competitive interactions [28].

Our data indicated that four similar-sized potential competitor species are not only sympatric with SJKF in the urban environment of Bakersfield, but that these species also commonly use all SJKF habitat suitability categories. Furthermore, all of the species extensively overlap spatially at specific locations based on detections at camera stations. This overlap further enhances the potential for either interference or exploitative competition between these species and SJKF. However, prior to the sarcoptic mange epidemic, SJKF were the second most frequently detected species at camera stations, with cats being the most frequently detected (see Figure 2). In 2015, which was 2 years after the epidemic began, SJKF detection rates were still at least four times higher than those of the other species except for cats. SJKF detection rates were also markedly higher than those of the other species in all three suitability categories. Therefore, it is unlikely that cats, skunks, raccoons, or opossums were limiting SJKF abundance. 

Conversely, SJKF does not appear to competitively exclude any of the other species. The significant reduction in SJKF abundance attributable to sarcoptic mange provided a natural experiment for assessing the effects of SJKF on other species. If SJKF had been competitively excluding any of the species, their populations would have increased as SJKF abundance declined (e.g., mesopredator release [35]). However, none of the other species exhibited an increase in detection rates in response to the SJKF population decline. Furthermore, a similar result was also observed in each of the SJKF habitat suitability categories. Under the conditions present during the 8-year study period, the population trends of SJKF and the other species were independent of each other. Finally, SJKF and the other species were commonly detected at the same camera stations. Therefore, the lack of association in population trends, overlap in use of habitat suitability categories, and co-occurrence at camera stations all provide evidence that SJKF and the other species are syntopically coexisting in the urban environment.

Several factors may facilitate the observed syntopic relationship between SJKF and the other species. None of these species constitutes a significant threat of interference or competition with each other. The adults of all of the species are similar in size. Donadio and Buskirk [13] found that in the absence of a size advantage between two potentially competing carnivores, the risk of injury in direct aggressive encounters is high, and therefore the best strategy is avoidance or tolerance. In multiple studies collectively involving several hundred radio-collared kit foxes in Bakersfield, no evidence has been found that any kit fox was killed by a cat, raccoon, skunk, or opossum [7]. Cat hair was detected in 1 of 900 kit fox scats [36,37], although whether this was a result of predation on an adult or on a kitten or a scavenging event is unknown. Raccoon, skunk, and opossum remains have not been detected in kit fox scats, although one observation of an adult kit fox killing a very young opossum was recorded during the camera station surveys.

Based on the many thousands of camera station images captured during this and other studies in the urban environment, aggressive interactions between SJKF and other species are almost non-existent. Even when a SJKF and another species are present simultaneously at a camera station, they appear to take turns investigating the attractant. SJKF have been observed to defer to other species, particularly skunks and large cats, but remain nearby. Similar interactions were observed at locations where food was being provisioned for feral cats but that were also visited by SJKF and skunks [38]. Thus, interference and competition between SJKF and the other species are apparently insignificant under current conditions, and this would facilitate syntopy between the species.

SJKF and the other species could potentially engage in exploitative competition, particularly for dens and food resources. SJKF are obligate den users. They use dens daily for daytime resting, evading predators, moisture conservation, avoiding temperature extremes, and rearing young [39,40]. They have multiple dens distributed throughout their home ranges and, on average, use a dozen or more different dens during the course of a year [41]. Thus, dens are a critical aspect of their natural history. Skunks, raccoons, opossums, and cats also use dens with varying frequencies. Skunks and opossums in the urban environment generally use dens during most days [38,42,43], while raccoons and cats use dens less consistently [44,45]. We have conducted multiple studies in which radio-collared SJKF in Bakersfield have been tracked to day-time resting locations, and the animals have almost always been found in a den. This indicates that den availability likely is not a limiting factor for SJKF and, therefore, presumably also is not a limiting factor for the other species.

Food availability in the urban environment may also not be a limiting factor for any of the species. SJKF in Bakersfield primarily consumes rodents (e.g., pocket gophers [*Thomomys bottae*] and California ground squirrels [*Otospermophilus beecheyi*]), birds, and a variety of invertebrates [36,37]. They also consume anthropogenically derived foods such as discarded fast food, food left out for pets, and food provided to feral cats [36,37,46]. Food habits for the other species have not been investigated in Bakersfield. However, based on opportunistic observations and data from other urban locations, the diets of these species likely overlap extensively with those of the SJKF [42,43,44,45,46,47,48]. Similar to SJKF, cats also commonly consume vertebrate and invertebrate prey. The other species have broader diets in that, in addition to animal prey, they commonly consume fruits and vegetable matter. All of the species also consume anthropogenic food, particularly cats, which commonly rely on food specifically supplied to them by humans.

The total combined availability of natural and anthropogenic foods in Bakersfield appears to be sufficiently high that food may not be a limiting resource for SJKF and the species. In particular, SJKF does not exhibit any signs of nutritional stress. Urban SJKF are commonly heavier on average than their counterparts in non-urban areas [49]. Urban SJKF also exhibits higher densities, survival, and reproductive success [6,7]. We are not aware of any information (e.g., observations of emaciated individuals) to suggest that food availability is limited for the other species. The ubiquity and abundance of anthropogenic foods throughout the urban environment is likely a primary factor. With abundant food resources and den sites in the urban environment, exploitative competition between SJKF and the other species is non-existent or insignificant under current conditions, and this would facilitate syntopy between the species.

Based on the detection rates for the species in each of the SJKF habitat suitability categories, the species exhibit some habitat partitioning. With the exception of raccoons, the use of the habitat suitability categories by the other species was disproportional to that of SJKF. Raccoons may have similar habitat preferences to SJKF, resulting in similar detection rates between the two species across habitat suitability categories. Competitive interactions among sympatric species are commonly assumed to result in niche partitioning, which also implies that the removal of a species would result in niche expansion by the remaining species [34,50]. However, as discussed, we found no evidence for significant interference or exploitative competition between SJKF and the other species. Also, the proportional use of the different habitat categories by the other species did not change over time in response to the significant decline in SJKF abundance due to sarcoptic mange. This indicates that differences in patterns of category use between SJKF and the other species were not a result of competition but instead likely reflected differences in habitat attribute preferences. Thus, the other species did not appear to be influencing habitat use by SJKF, and SJKF did not appear to be influencing habitat use by the other species. This provides further evidence that SJKF and the other species were coexisting syntopically in the urban environment.

## 5. Conclusions

Striped skunks, raccoons, opossums, and cats are all relatively common and widespread in the urban environment in Bakersfield, CA, and are potential competitors with the endangered SJKF. However, a combination of differences in ecological attributes (e.g., food and habitat preferences), reduced aggression due to size similarity and concomitant high injury risk, and an abundance of critical resources (e.g., food, dens) apparently sufficiently mitigate competitive interactions between SJKF and the other species, resulting in syntopic coexistence. This contributes to maintaining a diverse urban mesopredator community in Bakersfield. Syntopy also results in a fortuitous situation for SJKF in that it facilitates their persistence in the urban environment, thereby providing an additional population that lowers extinction risk and contributes to conservation and recovery efforts [5,6,7].

Our data and analyses provide substantial evidence of syntopy between SJKF and co-occurring mesocarnivore species, which are on our study site. Nevertheless, additional research on this question is warranted. In particular, the use of occupancy modeling would provide a more robust analysis of this question as well as potentially provide additional information on interactions between the species. Such an analysis would require that multiple surveys be conducted during a given time interval where occupancy patterns were stable. Although we did not have the resources to conduct multiple surveys per year, future efforts could include such a study design.

## Figures and Tables

**Figure 1 animals-13-03210-f001:**
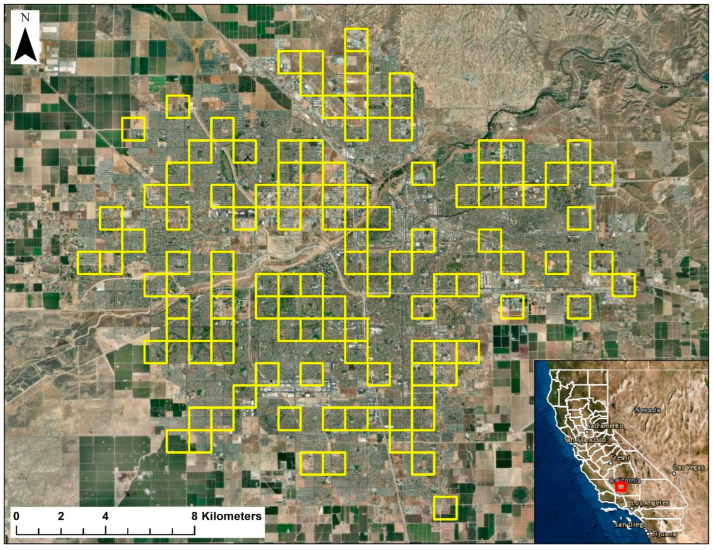
City of Bakersfield in Kern County, CA. The 1-km^2^ grid cells used to conduct annual camera station surveys are outlined in yellow.

**Figure 2 animals-13-03210-f002:**
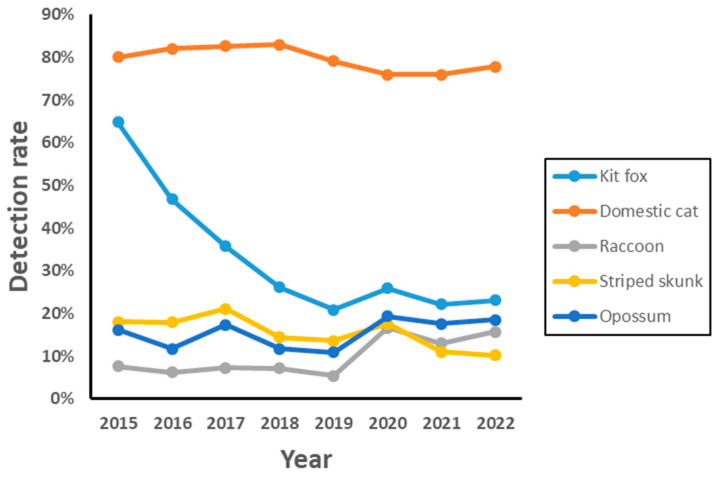
Annual detection rates for five species during 2015–2022 in Bakersfield, CA.

**Figure 3 animals-13-03210-f003:**
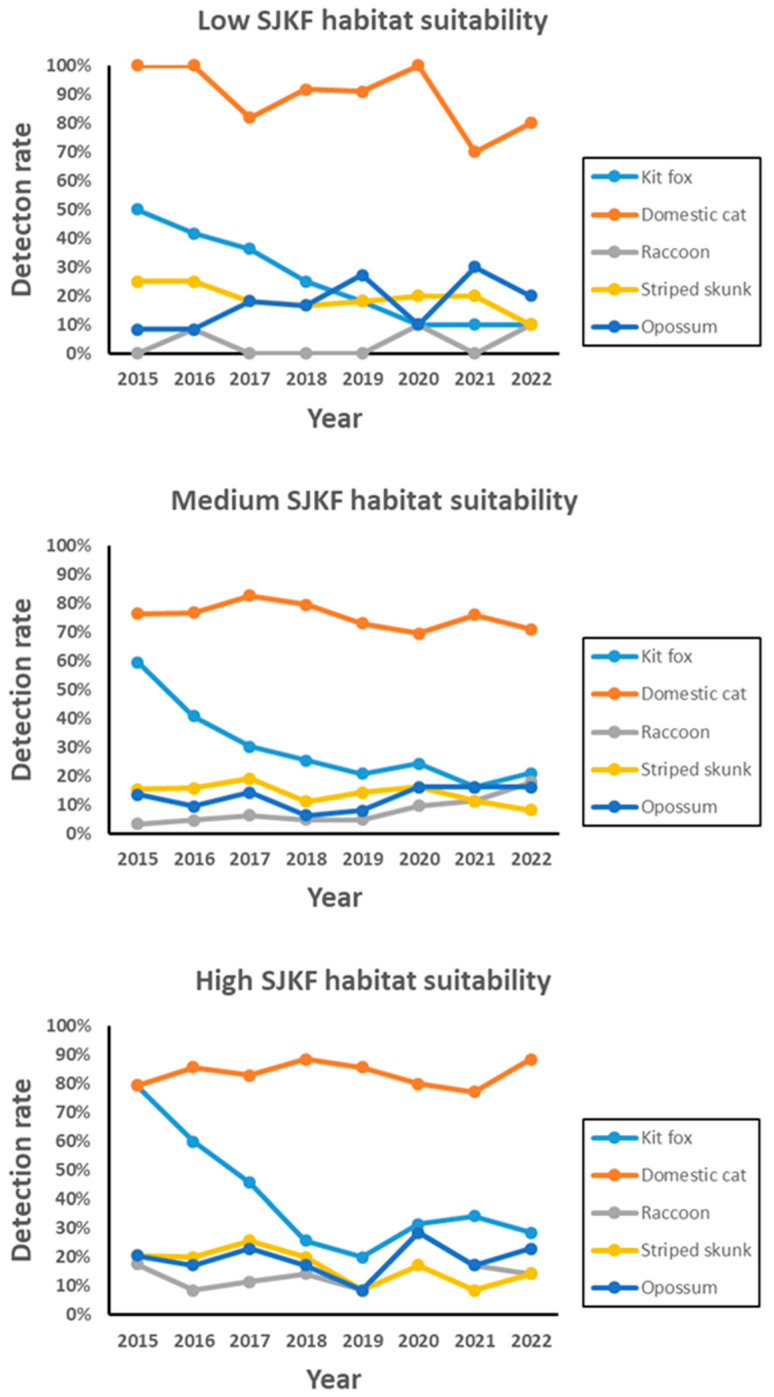
Annual detection rates for five species by SJKF habitat suitability category during 2015–2022 in Bakersfield, CA.

**Figure 4 animals-13-03210-f004:**
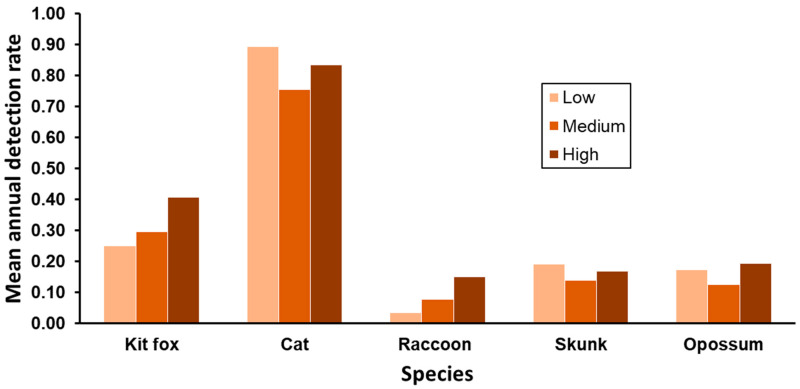
Mean annual detection rates by SJKF habitat suitability category for five species during 2015–2022 in Bakersfield, CA.

**Table 1 animals-13-03210-t001:** Mean annual detection rates (mean annual proportion of stations with detections) for five species during camera station surveys conducted during 2015–2022 in Bakersfield, CA.

Species	Detection Rate (%)				Correlation with SJKF	
	Mean	SE	Min.	Max.	*r*	*p*
Kit fox	33.2	5.5	20.9	64.8	-	-
Domestic cat	79.5	1.0	75.9	82.9	0.03	0.33
Striped skunk	15.5	1.3	10.2	21.1	0.28	0.10
Opossum	15.4	1.2	10.9	19.4	−0.16	0.92
Raccoon	9.9	1.6	5.5	16.7	0.01	0.34

**Table 2 animals-13-03210-t002:** The mean annual detection rates (mean annual proportion of stations with detections) for five species by SJKF habitat suitability category during camera station surveys conducted during 2015–2022 in Bakersfield, CA.

Species	Mean (±SE) Annual Detection Rate (%) by Habitat Suitability Category		
	Low	Medium	High
SJKF	25.1 ± 5.6 A ^1^	29.6 ± 5.0 A	40.6 ± 7.1 B
Domestic cat	89.4 ± 3.9 A	75.5 ± 1.2 B	83.6 ± 1.6 AB
Striped skunk	19.1 ± 1.7 A	13.8 ± 1.3 B	17.0 ± 2.1 AB
Opossum	17.3 ± 3.0 AB	12.4 ± 1.4 A	19.5 ± 2.1 B
Raccoon	3.5 ± 1.7 A	7.9 ± 1.7 B	15.1 ± 2.3 C

^1^ For each species, means with the same letter did not differ statistically.

**Table 3 animals-13-03210-t003:** Mean annual detection rates for four other species at camera stations with SJKF detections by SJKF habitat suitability category during surveys conducted during 2015–2022 in Bakersfield, CA.

Species	Mean (±SE) Annual Detection Rate (%) in Cells with SJKF by Habitat Suitability Category		
	Low	Medium	High
Domestic cat	90.6 ± 6.6 A	58.2 ± 6.0 B	76.2 ± 1.9 B
Striped skunk	6.7 ± 4.5 A	12.8 ± 1.7 B	15.9 ± 3.4 C
Opossum	0.0 ± 0.0 A	12.1 ± 2.4 B	20.1 ± 2.3 C
Raccoon	15.0 ± 12.4 A	5.2 ± 1.8 A	14.9 ± 3.1 A
All species	90.6 ± 18.7 A	65.7 ± 14.6 B	86.0 ± 9.0 A

For each species, means with the same letter did not differ significantly at α = 0.05.

## Data Availability

Data will be made available upon reasonable request.

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
