# Peer review of "Syntopy between Endangered San Joaquin Kit Foxes and Potential Competitors in an Urban Environment"

_animals, 2023, doi:10.3390/ani13203210_

Round 1
Reviewer 1 Report
For authors and editor
General comments:
In this study authors aim to assess whether competition occurs, or not, between the endangered San Joaquin kit fox and four other carnivores co-occurring in the city of Bakersfield. Data were collected from 2015 to 2022 following a city-wide camera station survey. Clearly this paper presents, and discuss, partial results of a wider study. The limitation with this option by the authors is that the discussion is not so strong as it could be if other complementary results were also included. When conducting research, it is crucial to consider and report relevant data and findings to provide a comprehensive and well-rounded analysis. For instance, including data on daily and seasonal variation in the activity of the different species can contribute to a more thorough understanding of their coexistence patterns and potential competition. But these data are missing for the 5 species of carnivores.
General appreciation:
The objectives of the study are clear and studies on the use of the urban environment by different species are always relevant under a conservation perspective. In addition, urban areas are expected to increase in the future and information is missing for its suitability for many species. However, although authors follow methodological procedures already tested in previous studies and the methodological steps are reasonably well described the study would greatly benefit from a richer discussion on the species-specific use of space throughout the year, influence of human disturbance (if occurring) throughout the year and especially conservation implications. For example, are there certain periods of the year when the SJKF is apparently more vulnerable (and in this sense is important to relate with the decreased population from 2015 to 2022), and conservation efforts might be most effective?
Other detailed comments:
Material and methods:
Several sentences need clarification. For instance:
Lines 142-143 – when considering the number of detections of each species how the authors deal with the fact that the same individual can be detected several times. Or not? Please clarify.
Results:
Figures 2 and 3 – The annual detection rates of opossums and raccoons slightly increase from 2019 onwards when the detection rate of kit foxes start to decrease (fig.2). This trend is also clear in medium and high SJKF habitat suitability (fig.3). Although there is no significant statistical correlation between these values authors should better comment these results in the discussion considering the ecological preferences of each species.
Figure 4 - The trend in the mean annual detection rates by habitat suitability category is similar for kit foxes and raccoons and as well similar for the other three species. Authors should better comment these results in the discussion considering the ecological preferences of each species.
Discussion:
Discussion must be improved.
Author Response
Response to Reviewer 1 comments
Manuscript animals-2621089
25 September 2023
We thank the reviewer for their thoughtful comments and suggestions.
Responses are in bold type
Comments and Suggestions for Authors
For authors and editor
General comments:
In this study authors aim to assess whether competition occurs, or not, between the endangered San Joaquin kit fox and four other carnivores co-occurring in the city of Bakersfield. Data were collected from 2015 to 2022 following a city-wide camera station survey. Clearly this paper presents, and discuss, partial results of a wider study. The limitation with this option by the authors is that the discussion is not so strong as it could be if other complementary results were also included. When conducting research, it is crucial to consider and report relevant data and findings to provide a comprehensive and well-rounded analysis. For instance, including data on daily and seasonal variation in the activity of the different species can contribute to a more thorough understanding of their coexistence patterns and potential competition. But these data are missing for the 5 species of carnivores.
General appreciation:
The objectives of the study are clear and studies on the use of the urban environment by different species are always relevant under a conservation perspective. In addition, urban areas are expected to increase in the future and information is missing for its suitability for many species. However, although authors follow methodological procedures already tested in previous studies and the methodological steps are reasonably well described the study would greatly benefit from a richer discussion on the species-specific use of space throughout the year, influence of human disturbance (if occurring) throughout the year and especially conservation implications. For example, are there certain periods of the year when the SJKF is apparently more vulnerable (and in this sense is important to relate with the decreased population from 2015 to 2022), and conservation efforts might be most effective?
The reviewer raises a valid point in the “General comments” and in the “General appreciation”. It would indeed be very desirable to have more information on what the species were doing year-round. Unfortunately, our surveys were only conducted during one season each year, that being summer. Our main purpose for the surveys was to document the incidence and distribution of kit foxes with sarcoptic mange. We also obtained information on other species, but our data essentially constitute a snapshot in time of the activities of these other species. Thus, we do not have any information on activities or behavior of the species during other seasons. This certainly could be the basis of further research, although just based on casual observations over the years and incidental camera data from stations where we searched for specific foxes with mange, our strong suspicion is that the activities and space use by the other species that we observed during our surveys was largely representative of other seasons as well. The urban environment is generally quite stable and does not experience the seasonal fluctuations in resource abundance and distribution that is observed in natural habitats. Consequently, the activities and responses by species in the urban environment do not appear to vary extensively.
As for kit fox vulnerability, we have not observed much variation in reports of mortalities among seasons or in our telemetry studies. The kit fox population decrease was indeed associated with the intensity of the mange epidemic. The kit fox detection rates were inversely related to the number cases of kit foxes with mange that were documented each year.
Other detailed comments:
Material and methods:
Several sentences need clarification. For instance:
Lines 142-143 – when considering the number of detections of each species how the authors deal with the fact that the same individual can be detected several times. Or not? Please clarify.
Also a good point. We were quite conservative in our estimates of the number of individuals. Individuals were identified based on sex, age and relative size, and any distinguishing marks or attributes, such as scars, old injuries, coloration differences, etc. Because of the uncertainty involved enumerating individuals, we did not use the counts of individuals in our analyses. We only considered whether a species was detected at a station or not.
Results:
Figures 2 and 3 – The annual detection rates of opossums and raccoons slightly increase from 2019 onwards when the detection rate of kit foxes start to decrease (fig.2). This trend is also clear in medium and high SJKF habitat suitability (fig.3). Although there is no significant statistical correlation between these values authors should better comment these results in the discussion considering the ecological preferences of each species.
The reviewer is indeed correct that opossums and raccoons exhibited a slight increase in the later survey years. However, we also noted that kit foxes increased concurrently in 2020, all declined slightly in 2021, and then all increased slightly again in 2022. So, it seemed apparent to us that the other species were not responding to kit fox abundance, but that all species were responding to some other factor that we were not able to identify. Also, the differences in detection rates for the species between those years was sufficiently small that we could not completely dismiss random variation.
Figure 4 - The trend in the mean annual detection rates by habitat suitability category is similar for kit foxes and raccoons and as well similar for the other three species. Authors should better comment these results in the discussion considering the ecological preferences of each species.
We added a statement in the Discussion acknowledging the similarity in detection rate patterns between kit foxes and raccoons across the habitat suitability categories and suggest that the two species may have similar habitat preferences in the urban environment.
Discussion:
Discussion must be improved.
We assume the reviewer was referring to the earlier comment regarding adding information about habitat preferences and responses for the other species during other seasons. As mentioned, we do not have that information due to the fact that surveys were only conducted in the summer.
Again, we would like to thank the reviewer for their thoughtful comments that helped improve our manuscript.
Reviewer 2 Report
Thank you for the opportunity to read the manuscript “Syntopy between endangered San Joaquin kit foxes and potential competitors in an urban environment”. This paper evaluated the co-occurrence of sympatric species with San Joaquin kit foxes and investigated whether habitat attributes influenced this co-occurrence. The manuscript was written very well. My main concerns, which are not what I would consider major concerns, are related to the general theme of literature sources. I think the introduction could benefit from additional focus on hypotheses that might explain the syntopic interactions that the authors are investigating. Furthermore, while I appreciate the recognition of landmark, foundational studies, it may add value to also incorporate newer, contemporary studies that could provide the reader with some modern, applied context for the research being discussed. Overall, I enjoyed this paper and wish the authors the best of luck with their revisions. Please see below for my specific comments and suggestions.
1. Line 71: Space is a likely mitigating factor for competing species, but what about time? I’m not suggesting that a temporal analysis be included, but maybe mention why time is not being addressed.
2. Lines 72-78: Can a hypothesis(es) that is/are being explicitly tested be provided? There has been a lot of work done on how kit foxes respond to competitors or predators, particularly considering habitat serving as a mediating factor diminishing or accentuating any effects. There has been a study with urban carnivores that may be relevant: Moll et al. 2018. Humans and urban development mediate the sympatry of competing carnivores. Urban Ecosystems 21:765-778.
3. Lines 128-129: I would imagine that kit foxes are more trap happy than trap shy, explaining why previous work has detected them within the first week of camera sampling. However, can the same be said for the other sympatric species that are included in this analysis? If not, the authors should acknowledge this as their findings may not have completely captured all other species’ activity.
4. Lines 220-221: Maybe clarify about the suitability categories in the figure caption. It could be interpreted that the suitability categories are for all the carnivores and not just the San Joaquin kit fox.
5. Line 246: the Birch 1957 citation is a great foundational piece of work but given the attention that interference and exploitative competition have received since then, it may add value to reader as they are assessing the discussion section to provide some more contemporary references that complement Birch 1957.
6. Lines 250-252: Great point! The data from prior years do not suggest a limiting effect from the other species.
Author Response
Response to Reviewer 2 comments
Manuscript animals-2621089
25 September 2023
We thank the reviewer for their thoughtful comments and suggestions.
Responses are in bold type
Comments and Suggestions for Authors
Thank you for the opportunity to read the manuscript “Syntopy between endangered San Joaquin kit foxes and potential competitors in an urban environment”. This paper evaluated the co-occurrence of sympatric species with San Joaquin kit foxes and investigated whether habitat attributes influenced this co-occurrence. The manuscript was written very well. My main concerns, which are not what I would consider major concerns, are related to the general theme of literature sources. I think the introduction could benefit from additional focus on hypotheses that might explain the syntopic interactions that the authors are investigating. Furthermore, while I appreciate the recognition of landmark, foundational studies, it may add value to also incorporate newer, contemporary studies that could provide the reader with some modern, applied context for the research being discussed. Overall, I enjoyed this paper and wish the authors the best of luck with their revisions. Please see below for my specific comments and suggestions.
See responses below that address the issues raised in this summary paragraph.
- Line 71: Space is a likely mitigating factor for competing species, but what about time? I’m not suggesting that a temporal analysis be included, but maybe mention why time is not being addressed.
Good point. We added some additional ways in which competition could be mitigated. And we actually have tried looking at temporal partitioning in another paper. However, I (senior author) have some concerns about regarding temporal analyses based on camera data. The cameras are very dependent upon an individual passing in front of the camera, and there are myriad reasons that they might not do so at a particular time. And the cameras have a limited field of view meaning that an individual could even be present in the immediate area but not detected. In general, my feeling is that camera data are too coarse for any sort of fine-scale temporal analysis, at least on a daily basis. In our study, multiple species frequently were detected on the same station on the same night, although the order of appearance varied.
- Lines 72-78: Can a hypothesis(es) that is/are being explicitly tested be provided? There has been a lot of work done on how kit foxes respond to competitors or predators, particularly considering habitat serving as a mediating factor diminishing or accentuating any effects. There has been a study with urban carnivores that may be relevant: Moll et al. 2018. Humans and urban development mediate the sympatry of competing carnivores. Urban Ecosystems 21:765-778.
Again, good suggestion. We added a statement describing the general hypothesis that we were testing. And thank you for the Moll et al. reference. We have added that to the paper as well.
- Lines 128-129: I would imagine that kit foxes are more trap happy than trap shy, explaining why previous work has detected them within the first week of camera sampling. However, can the same be said for the other sympatric species that are included in this analysis? If not, the authors should acknowledge this as their findings may not have completely captured all other species’ activity.
We added a statement regarding this. We do not have quantitative information on time to detection for the other species, although in examining images from the cameras over the many years of the survey, we felt confident that seven nights was sufficient to detect a species if it was present near a given camera location.
- Lines 220-221: Maybe clarify about the suitability categories in the figure caption. It could be interpreted that the suitability categories are for all the carnivores and not just the San Joaquin kit fox.
Good catch. We added clarification to the caption.
- Line 246: the Birch 1957 citation is a great foundational piece of work but given the attention that interference and exploitative competition have received since then, it may add value to reader as they are assessing the discussion section to provide some more contemporary references that complement Birch 1957.
This is a good point. We added a new paragraph at the beginning of the discussion that hopefully provides a bit more context to the question being addressed and that cites a number of newer works relevant to our study.
- Lines 250-252: Great point! The data from prior years do not suggest a limiting effect from the other species.
Again, we thank the reviewer for excellent comments and suggestions that helped to improve our manuscript.
Reviewer 3 Report
see attached

Author Response
Response to Reviewer 3 comments
Manuscript animals-2621089
25 September 2023
We thank the reviewer for their many thoughtful comments and suggestions.
Responses are in bold type
General comments
This paper uses existing data from a camera trap survey in the US town of Bakersfield to explore detection rates of San Joaquin kit foxes (SJKF). The data collection method was baited camera traps set-up in a grid and recoding data for 7 consecutive days. These data are then compared to detection rather for other similar sized exotic and native species (cat, racoon, skunk and possum) in what are classified as low, medium and high quality SJKF habitat within the study area.
The paper is very well written with very few grammatical or typographic errors. The camera trap array is an appropriate method for sampling medium sized animals in a mixed-use landscape. My main concern is the use of detection rates as the primary response variable. Ecological communities are composed of multiple interacting species. Better methods have been developed to improve our ability to draw inference from such communities by permitting modelling of detection/non-detection data. For example, multispecies occupancy models for two or more interacting species permits modelling the probability that two or more species occur together as a function of environmental variables. These advancements represent an important improvement in our ability to draw community-level inference from multiple interacting species that are subject to imperfect detection. By using detection rates for species that differ in detectability and probably the likelihood to explore the bait provided, the authors have introduced biases to their data. This does not make the analyses wrong only not as robust as they might have been had they used these more advanced and recent modelling approaches.
We considered employing occupancy modeling to analyze our data and even consulted a biometrician. However, we elected not to use this approach because our data violate several important assumptions necessary to conduct valid occupancy modeling. First, we only conducted one survey per year. Occupancy modeling requires multiple surveys per time period. We might have considered defining the time period as the entire 9-year study period with each annual survey being a constituent survey. However, a basic assumption of the modeling is that the probability of occupancy of a given species remain consistent or vary randomly during the time period. Otherwise, the assumption of “closure” is violated. Clearly, occupancy probability varied considerably between years, if for no other reason than the kit fox population declined significantly due to the mange epidemic. Also, if all of the annual surveys had been used in the manner above, then we would have only had a single time period.
Another issue was that occupancy models are notoriously “data hungry”. In order to incorporate covariates for detection, covariates for occupancy, and do so for the 3 habitat suitability categories, 5 species, and (for some analyses) stations with and without kit fox detections, we would have needed considerably larger sample sizes.
So again, we completely agree with the reviewer that multi-species occupancy modeling is a powerful analytical approach given that appropriate data are available. However, we did not feel that our data met the assumptions for this technique and therefore were not appropriate.
A second concern is the disease outbreak in the SJKF which resulted in a strong negative impact on the detection rate of this species. The authors use this disease to provide a quasi-experimental element suggesting that the absence of a change in detection rates of other species while the fox rate declined is evidence that they are not competing species. While this may be the case, I do not think the data nor the analyses used are sufficiently robust to allow for this conclusion to be reached. Multispecies occupancy models (e.g., Rota et al. 2016) allow for pairwise interactions among most species, and the probability of some pairs of species occupying the same site may well vary along environmental gradients. Thus, for example, occupancy probabilities of two species could be independent at sites with little human disturbance, but in areas of high human disturbance the same two species might be more likely to occur together. This is a much more meaningful way of exploring interactions between species in a community while simultaneously factoring in environmental covariates.
Again, we agree with the reviewer that multispecies occupancy models are a powerful tool for exploring species interactions, assuming that appropriate and sufficient data are available. For the reasons detailed previously, we did not feel that our data were appropriate or sufficient to employ this approach. Thus, we were limited to other approaches. As for the effects of the temporal change in kit fox abundance, the population declined by about two-thirds during the study. This is a very significant reduction in abundance. Based on our familiarity with the species and their ecology, we felt quite confident that with a decline of that magnitude, if kit foxes were indeed competitively suppressing other species, then even with the analytical approaches we employed, we would have seen some significant positive response by those species. We assessed this in a couple of different ways and did not detect such a response. Thus, we have a high degree of confidence in our conclusion that the other species did not exhibit a detectable response to the decline in kit fox abundance.
In summary the authors have provided a simple subset of data from an existing data set in the same study site and using the same camera trap array. These data have already been used to explore occupancy models of SJKF (Deathridge et al. 2021) and occupancy is a preferred metric to detection rates. These data have also been used to explore the extent of the mange outbreak in a recent publication (Kelly et al. 2022). In the latter paper occurrence data is also provided for multiple years for kit foxes and all of the species in this study (racoon, opossum, skunk, domestic cat) and many others too. This paper attempts to distinguish itself from these previous two publications by comparing detection trends in low, medium and high quality habitat but as suggested above the analyses used are not suitable for a rigorous assessment of this question.
This paper is indeed considerably different from the others cited above by the reviewer. The Deatherage et al. 2021 paper addressed a very straightforward question. It explored the habitat attributes associated with the occurrence of kit foxes. It did not address interactions between other species co-occurring with kit foxes in the urban environment. The Kelly et al. 2022 paper also addressed a very straightforward question. It simply documented the presence and relative frequency of detection of other species (most of which were not addressed in the current study) at the camera stations. It did not attempt to address potential competitive interactions between species. The goal of that paper was to identify other species that could potentially be involved in the transmission of mange. Statements in that paper actually support well the conclusions in the current manuscript because many of the species were detected at the same stations where kit foxes were detected.
Specific comments 1
97-98: To assess syntopy between SJKF and other species, we used occurrence data gathered from 2015 to 2022 during annual surveys conducted in Bakersfield to assess the frequency and distribution of SJKF with sarcoptic mange [17].
Grammar – suggest rewrite as follows: we used occurrence data gathered from 2015 to 2022 during annual surveys conducted in Bakersfield to assess the frequency and distribution of SJKF with sarcoptic mange [17]. These data were used to infer syntopy between SJKF and other species.
However, the meaning of this is still unclear. Are you saying that you index of syntopy is based on SJKF with mange?
We implemented the text change suggested by the reviewer. Syntopy was based on all kit fox detections. Only a relatively small percentage of the foxes detected on camera exhibited signs of mange.
103-104: We used the randomization function in Excel 2010 (Microsoft, Redmond WA) to randomly select grid cells to potentially sample from the 357 available cells.
Suggest rewrite: We used the randomization function in Excel 2010 (Microsoft, Redmond WA) to randomly select a subsample of grid cells from the 357 available cells.
We revised this text.
105-106: Usually, 111 grid cells (31%) were sampled annually, and the same cells were sampled each year to eliminate inter-annual spatial variation as a confounding variable (Figure 1).
Suggest rewrite: This resulted in a sample of 111 grid cells (31% of the total) which were then sampled annually (Figure 1).
We revised this text.
We at-112 tempted to find locations toward the center of cells in which to place cameras such that 113
stations in adjacent cells would be separated by at least 1 km.
What was the reasoning behind a 1km spacing between camera traps? Usually spacing is linked to the home range(s) of the study species with the goal of sampling stations being independent of on another. If this was the goal then we need home range data for the species to assess how well the assumption of independence is being met.
We provided the estimated home range size for urban SJKFs to clarify the rationale for the placement of the camera stations.
Because 97.1% of the first detections of a SJKF at camera stations typi-128 cally occur within six nights [18],
I cannot access this paper so please specify whether the study referred to in [18] was for a baited station?
The stations entailed use of an attractant. The referenced paper is available at:
Again, we thank the reviewer for the helpful and insightful comments.
Round 2
Reviewer 3 Report
The authors have argued their points well and conceded that more data (i.e., a longer camera trap survey) would have allowed for better analyses (e.g., MSOM) to understand how species are influenced by each other and environmental variables. While I think the design is weak for the question asked that is decision for the editor not the reviewer as to whether it is publishable. I accept that they use word 'infer' when referring to syntopy in the species present and this is appropriately circumspect. I would however like to recommend that the authors signal to the readership in the introduction and discussion what the weaknesses of their approach are and how they could be addressed to better approach the question of syntopy or sympatry in multiple species.
Author Response
We again would like to thank the reviewer for their interest in our paper and the helpful comments. In response, we felt that an assessment of the strengths and weaknesses of the analysis we conducted would be best presented at the end of the manuscript after all of our information had been presented. To that end, we included a new paragraph at the end of the Conclusions that recommends additional research into the question of syntopy among urban mesocarnivores and that also recommends the use of a more robust study design and analytical methods. We hope that adding this information addresses the concern of the reviewer. Thanks again for the comments.